# Potential of Chitosan for the Control of Powdery Mildew (*Leveillula taurica* (Lév.) Arnaud) in a Jalapeño Pepper (*Capsicum annuum* L.) Cultivar

**DOI:** 10.3390/plants13070915

**Published:** 2024-03-22

**Authors:** Omar Jiménez-Pérez, Gabriel Gallegos-Morales, Cesar Alejandro Espinoza-Ahumada, Carolina Delgado-Luna, Pablo Preciado-Rangel, Bernardo Espinosa-Palomeque

**Affiliations:** 1Departamento de Parasitología, Universidad Autónoma Agraria Antonio Narro, Calzada Antonio Narro, Saltillo 25315, Mexico; 2Agricultura Sustentable y Protegida, Universidad Tecnológica de Escuinapa, Camino al Guasimal S/N, Colonia Centro, Escuinapa de Hidalgo 82400, Mexico; bespinosa@utescuinapa.edu.mx; 3Departamento de Ingeniería en Innovación Agrícola, Instituto Tecnológico Superior de El Mante, Km 6.7, México 85, Quintero 89930, Mexico; caespinoza@itsmante.edu.mx; 4Campo Experimental Río Bravo-INIFAP, Km 61 Carretera Matamoros, Río Bravo 88900, Mexico; delgadolunac29@gmail.com; 5Tecnológico Nacional de México, Instituto Tecnológico de Torreón, Carretera Torreón-San Pedro Km 7.5, Torreón 27170, Mexico

**Keywords:** agrochemicals, biopolymer, powdery mildew, control efficacy

## Abstract

One of the phytopathogens that cause severe damage to jalapeño pepper is *Leveillula taurica* (Lév.) Arnaud, the causative agent of powdery mildew. Synthetic fungicides are currently employed for its control, contributing to adverse effects on human health and the environment. The main objective of this research was to identify the causal agent of powdery mildew and assess the efficacy of chitosan in powdery mildew control on jalapeño pepper. The following treatments were evaluated in laboratory and greenhouse conditions: T1 = 0.0125% chitosan, T2 = 0.0025% chitosan, T3 = 0.05% chitosan, T4 = 0.1% chitosan, T5 = 0.2% chitosan, T6 = tebuconazole 25% (1.8 mL/L water), and T7 = control (water). Symptomatology results indicated that *L. taurica* is indeed the causative agent of powdery mildew. Treatments T4 and T5 exhibited the lowest percentages of incidences and severity, hence achieving higher control efficacy in the laboratory (57.70 ± 3.85 and 65.39 ± 3.85) and greenhouse (56.67 ± 4.08 and 70 ± 8.16%) compared to T6 (control efficacy, 38.46 ± 0.00% in the laboratory and 50 ± 0.00% in the greenhouse). The chitosan derived from shrimp had a significant impact on the cell walls of *L*. *taurica* spores and mycelium. Consequently, chitosan emerges as a viable organic alternative to fungicides for controlling powdery mildew in jalapeño pepper.

## 1. Introduction

The *Capsicum* genus encompasses over 200 species, exhibiting a wide spectrum of color, size, shape, and chemical composition. The fruits of the jalapeño pepper (*Capsicum annuum* L.) have diverse applications, being consumed fresh, dried, in pastes, and incorporated into sauces, pickles, or smoked foods [1]. Powdery mildew disease, attributed to *Leveillula taurica* (Lév.) Arnaud (Anamorph: *Oidiopsis taurica* (Lév.) Salmon) [2], holds substantial significance for chili cultivation. This cosmopolitan phytopathogenic fungus affects more than 1000 plant species, including members of Compositae and Leguminosae. However, the most crucial host crops belong to Solanaceae [3,4]. In Mexico, this phytopathogen was first identified in 1979 in the state of Sinaloa, impacting tomato crops (*Solanum lycopersicum*) [5]. Powdery mildew significantly affects the leaves of jalapeño pepper grown both in greenhouses and fields, leading to reduced yield and compromised fruit quality [6]. Estimated crop yield losses due to powdery mildew can reach up to 31% [7].

The germination of powdery mildew conidia occurs at temperatures ranging from 10 to 37 °C, with optimal germination observed at 20 °C, decreasing at 40 °C for 6 h. Leaf colonization is optimal between 15 and 25 °C, with increased leaf infections between 15 and 20 °C and suppressed infections between 20 and 25 °C. Higher germination rates are observed at relative humidity levels between 75 and 85%. The highest foliar coverage of the disease is prolonged between 10 and 15 °C, coupled with diurnal relative humidity between 85 and 95% [8]. Symptoms on plants are manifested through the presence of mycelium and conidiophores emerging through stomata on the underside of leaves, forming whitish powdery spots. On the upper side, these spots exhibit a yellowish to brown coloration, progressing from lower to upper parts of the plants. As the disease advances, symptoms intensify, leading to premature defoliation and adversely affecting crop production [9].

Synthetic fungicides such as myclobutanil, triflumizole, pyraclostrobin + boscalid, quinoxyfen, and strobilurin derivatives (azoxystrobin, trifloxystrobin, and kresoxim-methyl) are applied to control powdery mildew in chili [4]. However, fungicide application leads to soil fertility deterioration, residual toxicity in fruits, development of resistance by plant pathogens, loss of biodiversity, and increased costs of plant protection [10]. Currently, there are alternative approaches for effectively controlling a diverse range of phytopathogens, with biological control being a notable option. This involves the application of antagonistic microorganisms, including *Bacillus* spp., *Pseudomonas fluorescens*, *Trichoderma* spp. [11,12,13,14], *Streptomyces* spp., and yeast species such as *Rhodotorula* minuta, *Aureobasidium pullulans*, *Candida azyma* [13], *C. oleophila*, and *C. saitoana* [11,12]. These microorganisms employ several mechanisms of action, such as the production of compounds toxic to fungi (e.g., peroxidase, 3-phenyl lactic acid, allyl phenylacetate, and 2-propenyl ester), generation of reactive oxygen species (ROS), induction of plant defense mechanisms, and competition for space and nutrients [13,14]. In addition to the application of antagonistic microorganisms, antifungal compounds extracted from organisms can be employed, such as plant compounds (oils and extracts) and chitosan, which have activity against a wide variety of phytopathogens, in addition to being nontoxic and biodegradable [14]. Chitosan is a biopolymer derived from the deacetylation of chitin, and has an antagonistic effect against various phytopathogens, including *L. taurica* in tomato plants [15]. In one study, after an induction phase of 3 days between the application of chitosan and the inoculation of the phytopathogen powdery mildew in barley (*Blumeria graminis* f. sp *hordei*), the infection in the primary leaf was significantly reduced by 55.5%, due to the fact that chitosan is an exogenous inducer of acquired systematic resistance, whose activity is due to its polycationic structure, and to the binding proteins of 78 kD [16].

Chitosan acts to enhance soil structure and has proven effective in reducing *Fusarium* wilt and damage caused by *Cylindrocladium floridanum*, *Alternaria solani*, and *Aspergillus flavus*. Its application extends to foliar administration, acting as a supplement in hydroponic solutions and an additive in plant tissue culture mediums. This versatility of use not only contributes to increased crop yields but also stimulates plants’ defensive systems, fostering overall plant growth [17]. The antagonistic capacity of chitosan arises from the interaction between the positive charges of the chitosan’s glucosamine amino group (+) and the negative charges of the fungal cell membrane [(phospholipids (−)]. This interaction affects the permeability of the cell membrane, inducing lysis and the subsequent release of cellular content. Additionally, chitosan exhibits multiple modes of action, including interference in mRNA synthesis and the chelation of essential elements crucial for microorganism development [18]. Furthermore, its application provides the benefit of fostering the development of beneficial microorganisms [19] and promoting comprehensive plant growth [20,21]. Therefore, the main objective of the current research was to identify the causal agent of powdery mildew and assess the efficacy of shrimp shell chitosan in controlling powdery mildew in the Mixteco F1 jalapeño pepper under laboratory and greenhouse conditions.

## 2. Results

### 2.1. Identification of the Causal Agent of Powdery Mildew on Jalapeño Pepper Plants

The morphological identification of the phytopathogen causing powdery mildew on jalapeño plants coincided with the characteristics described for the species *L*. *taurica*. Whitish powdery mildew, consisting of hyaline, septate mycelium and conidiophores (Figure 1A), was observed on the underside of the leaves and on the upper side of the leaves there were yellowish chlorotic spots (Figure 1B). Conidiophores emerging from leaf stomata (Figure 1C) were simple, with two to three branches (Figure 1D) from which conidia of lanceolate (Figure 1E(a),F) and cylindrical (Figure 1E(b),F) types were produced. These characteristics are similar to those reported by Salmon [22], Mosquera et al. [23] and Glawe [24]. The conidia presented the following dimensions: 70 µm long × 18 µm wide for the lanceolate form and 65 µm long × 18 µm wide for the cylindrical form, similar to measures previously reported for *L. taurica* [25,26]. In the present study, no sexual spores (cleistothecia) were observed on infected leaves, as also reported by Jones et al. [27] and Abdel-Azeem and Abdel-Moneim [28] who did not observe these structures.

### 2.2. Control of Leveillula taurica on Pepper Leaves by Application of Chitosan under Laboratory Conditions

#### Experiment 1

At the beginning of the experiment, plants with a powdery mildew severity percentage of 57% were selected for application of the evaluated treatments. In this study, chitosan acted as a biocontrol substance, with each evaluated treatment resulting in significant differences in the variables of severity, incidence, and control efficacy on powdery mildew on jalapeño pepper leaves (*p* ≤ 0.05). The results show that the application of treatment T3 (0.05% chitosan) does not reduce the severity of powdery mildew (Figure 2A); however, at higher concentrations, a decrease in the severity of powdery mildew was observed. The application of treatment T4 (0.1% chitosan) significantly decreased the severity at 24 h (53.57 ± 6.84%), 96 h (39.29 ± 3.57%) and 186 h (39.29 ± 3.57%). The control of cenicilla with treatment T5 (0.2% chitosan) was effective, with reductions at 24 h (50 ± 4.12%), 96 h (32.14 ± 3.57%) and 186 h (32.14 ± 3.57%). The value of control efficacy at 24 h was higher than that recorded with treatment T6 (tebuconazole 25%) (24 h, 42.86 ± 0.00%), and after 96 h the control was not effective with the T6 treatment because the severity had increased (96 h, 53.57 ± 3.57% and 186 h, 57.14 ± 0.00%) showing higher values than those presented with the T4 and T5 treatments at 96 h and 186 h. It was observed that the T6 treatment lost effectiveness in the control of powdery mildew after 24 h. Fungicide application can control the disease, but the efficacy depends on early detection and exhaustive coverage of the plant, which can be difficult. There were significant differences for the T4 and T5 treatments, registering higher percentages in the control efficacy variable after 96 h compared to the rest of the treatments (*p* ≤ 0.05). In general, the treatments with chitosan showed better control of powdery mildew compared to treatment T7 (control), with treatment T5 being the one that showed the highest control efficacy (65.39 ± 3.85%), which indicates the efficacy of chitosan in controlling powdery mildew (Figure 2B).

### 2.3. Chitosan Activity against Leveillula taurica in Greenhouse Conditions

#### Experiment 2

The results obtained from cultivating jalapeño pepper under greenhouse conditions closely resembled those obtained in the laboratory. Elevating the concentration of chitosan revealed significant differences in the variables of incidence (Figure 3), severity (Figure 4), and the percentage of control efficacy (*p* ≤ 0.05) (Figure 5). The application of T5 resulted in the lowest incidence (4.55 ± 2.88%) and the lowest severity (25.72 ± 7.00%), which in turn was reflected in better control efficacy (70.00 ± 8.16%), which was statistically equal to treatment T4 and superior to the rest of the treatments with chitosan and to treatment T6 (control efficacy 50 ± 0.0%). Treatment T7 (control) resulted in an incidence of 94.29 ± 5.71% and a severity of 85.71 ± 0.00% of powdery mildew. Figure 6 shows how treatment T5 resulted in no symptoms or signs of powdery mildew (Figure 6E), while treatment T7 showed symptoms and signs of the disease caused by *L. taurica* (Figure 6G).

## 3. Discussion

The morphological characteristics of powdery mildew observed on chili leaves are characteristic of *L*. *taurica*, such as its oval, lanceolate-shaped, unicellular conidia, which distinguishes it from other diseases caused by powdery mildew such as those reported by Mosquera et al. [23], Zheng et al. [29] and Romero [30].

In the literature, there is limited information regarding the efficacy of chitosan in controlling *L. taurica* in jalapeño pepper plants. Dafermos et al. [15] reported that the application of 0.5 g/L of chitosan to tomato plants reduced the progression of *L*. *taurica-*induced disease from 31.4% to 9.1%, compared to a control treatment in which it decreased from 34.1% to 12.9%. In comparison, the chitosan concentration of 0.2% utilized in our study exhibited a remarkable control efficacy of up to 70.00 ± 8.16% when compared to the control treatment.

According to Arici and ÖZkaya [31], the control of powdery mildew by applying fluopyram + tebuconazole to canyon F1 chili bell pepper crops in greenhouse conditions obtained 87.00 ± 0.39% control efficacy and 3.30 ± 0.13% reduction of severity. Similarly, the application of Timorex Gold^®^ (tea tree oil) obtained 6.00 ± 0.14% reduction of severity and 80.0 ± 0.45% control efficacy, values that were higher than those recorded in the present study in treatment T6, which had 50.00 ± 0.00% control efficacy and 42.86 ± 0.00% reduction of severity. In the present study, the T5 treatment registered the highest control efficacy (70.00 ± 8.16%) compared to the treatments with fluopyram + tebuconazole and Timorex Gold^®^, demonstrating that chitosan has better control efficacy on powdery mildew in comparison with synthetic fungicides and organic extracts.

The use of chitosan as a preventive treatment at a concentration of 100 mM in the cultivation of bell pepper cv., California grown in greenhouse conditions, was able to reduce the severity of powdery mildew, registering values of 10.00% and 10.60%, compared to the control showing 82.10% and 84.00% reduction of severity in two crop cycles, respectively [32]. The application of chitosan with other substances such as saccharin, calcium chloride, *Saccharomyces cerevisiae*, and potassium mono dihydrogen phosphate significantly reduced the incidence of diseases such as powdery mildew, downy mildew, and early blight and late blight in cucumber, melon, chili and tomato crops [33].

In addition, Domínguez-Serrano et al. [34] report that the application of 0.013% of chitosan for the control of *Podosphaera pannosa* reduced the incidence of powdery mildew by 60.3% and 63.1%, being more effective than that reported in this study at a similar concentration of chitosan (34.62% control efficacy), possibly because they are two different genera of powdery mildew and therefore have different tolerance to this biopolymer. This is because chitosan is an antitranspirant and is used to protect plants against oxidative stress, stimulating plant growth. Furthermore, chitosan is a natural, low-toxicity, inexpensive, biodegradable, and environmentally friendly product with diverse functions in agriculture. The foliar application of chitosan enhances plant growth and enhances the quality of jalapeño pepper fruit. One of the roles of chitosan is to mitigate water stress, which adversely impacts plant productivity. This is achieved by augmenting stomata conductance, net photosynthetic activity, and CO_2_ fixation, and concurrently reducing transpiration to conserve water in agricultural crops [32].

Chitosan acts in several ways against phytopathogens; for example, it stimulates defense mechanisms in different crops, including solanaceae; induces local or systemic acquired resistance; stimulates the synthesis of salicylic acid, phytoalexins, and pathogenesis-related proteins (chitinases and β-1,3-glucanases), which degrade the fungal cell wall; and stimulates callose production and lignification of plant cell walls through the production and accumulation of phenolic compounds (hydroxycinnamic acid (coumaric, caffeic, and feluric) and benzoic acid (benzoic, pyrotocatechuic, and gallic) [35]. Increased production of phenolic compounds is activated through the phenylpropanoid pathway, which increases tolerance to phytopathogens. In the bell pepper crop, it induces increased production of chitinase (EC 3.2.1.14), β-1,3-glucanase (EC 3.2.1.39) [36], catalases (cat1), phenylalanine ammonia lyase (pal) and pathogenesis-related protein 1 (pr1) genes, important genes in the response against stress caused by biotic and abiotic factors [37].

The mode of action of the biopolymer chitosan as a biocontrol is due to the union, through its positive charges (glucosamine amino groups), with the negative charges of the phytopathogens (phospholipids). This causes cell lysis [18,36,37,38,39,40], which generates an ionic homeostatic imbalance of K^+^ and Ca^2+^ and causes the exit of molecules (phosphates, nucleotides, and substrates of enzyme reactions), as a result of which respiration is affected. Inside the cell, chitosan exhibits electrostatic attraction to the negative charges of the phosphates of the nucleic acid chain, which affects DNA in the synthesis of mRNA and prevents protein synthesis [41]. In the mycelium, it causes nutrient loss and vacuolation, thin, distorted hyphae and malformations [42]. In spores, the loss of their cellular content and the rupture of the vacuole has been reported [43], and the latter could be related to the observed effect of dehydration of *L*. *taurica* spores after application of chitosan (Figure 7A). Tebuconazole inhibits the biosynthesis of ergosterol, a key component of fungal cell membranes [44], so the visualization of spores under the microscope after the tebuconazole treatment showed similar characteristics to those observed after the application of chitosan treatments (Figure 7B), while in the T7 treatment the spores were not observed to be dehydrated (Figure 7C). In addition, the application of this biopolymer also promotes plant growth; in potato, it is reported that it stimulates stomatal closure and reduces the effects of abiotic stress. In chili bell pepper plants, it stimulates germination, seedling emergence, and increases height, stem diameter, biomass and yield compared to controls [45]. Similarly, Reyes-Perez et al. [46] mentioned that in tomatoes, germination, development and production increased due to the fact that chitosan increases photosynthesis and chlorophyll production, helps in stomatal closure, which makes water use more efficient, and its decomposition can produce ammonium, which is used by the plant.

The results found in the present study strengthen the hypothesis that the application of chitosan is an organic alternative to the use of synthetic fungicides for the control of powdery mildew caused by *L*. *taurica* in jalapeño pepper.

## 4. Materials and Methods

### 4.1. Location of the Study

This study was carried out at the Antonio Narro Agricultural Autonomous University (UAAAN), in the microbiology laboratory and greenhouse belonging to the Department of Agricultural Parasitology, located in the city of Saltillo, Coahuila, México, at the geographical coordinates 25°21′13″ N latitude and 101°1′56″ W longitude, with an altitude of 1742 masl.

### 4.2. Morphological Identification of the Causal Agent of Powdery Mildew in Jalapeño Pepper

Mixteco F1 jalapeño pepper plants, 50 days after germination, were transplanted into 1 L unicel cups and placed in a greenhouse. The plants were visually monitored to observe the symptoms and signs of powdery mildew disease. Once the disease appeared (30 days after transplanting), plants were homogenized with the same percentage of incidence and randomly distributed for the identification of the causal agent of the disease and subsequently to carry out the evaluated treatments.

For the identification of powdery mildew, 10 leaves were collected from plants with symptoms and signs of the disease (whitish powdery mildew on the underside of the leaf). The leaves were taken to the laboratory, where microscopic preparations were made with lactophenol blue to be observed under a compound microscope (Motic BA210E) at 40× magnification and a scanning electron microscope (Hitachi mod. 3000). The morphological characteristics of the fungus were compared with those described by Salmon [22] and Mosquera et al. [23].

### 4.3. Evaluation of the Control of Powdery Mildew on Jalapeño Pepper Leaves under Laboratory Conditions by the Application of Chitosan

#### Experiment 1

Jalapeño pepper leaves that presented visual symptoms of the disease were collected and homogenized, with the help of a 7-level visual scale, to a level of 4, corresponding to a severity of 16 to 35% (Figure 8). These leaves were placed individually, with the petiole wrapped in moist cotton inside plastic containers (Clear Food Container; 4.25″ long × 4.25″ wide × 2.5″ deep) to avoid dehydration. For the treatments, chitosan extracted from shrimp shells with characteristics of 100% deacetylation and a viscosimetric molecular weight (Mv) of 457,000 g-mol^−1^, belonging to the Microbiology Laboratory of the UAAAN, was used. This chitosan was diluted in sterile water acidified with acetic acid at pH 5.5 to obtain the required percentages. A completely randomized design with the following seven treatments was used: T1 = 0.0125% chitosan, T2 = 0.0025% chitosan, T3 = 0.05% chitosan, T4 = 0.1% chitosan, T5 = 0.2% chitosan, T6 = tebuconazole 25% (1.8 mL/L water), and T7 = control (water), with four replicates, which were kept in the laboratory at a temperature of 28 °C and a photoperiod of 12:12 light: dark until the end of the experiment. The concentrations of chitosan in the present study ranged from 0.0125 to 0.2%, because in a previous study the application of 0.013% chitosan decreased the incidence of *Podosphaera pannosa* by 31.2% (February–April) and 19.3% (May–July), and applications of 0.025 to 0.2% reduced the development of powdery mildew in rose by 43.5 to 85.4% [34].

Treatments were applied by spraying with a 100 mL manual sprayer, applying approximately 1.5 mL of solution per leaf, after 24, 96 and 186 h. Disease severity was evaluated with the help of the 7-level visual severity scale (Figure 8) at 24 h after each treatment application. The data obtained with the scale were transformed to severity percentages using the Formula (1) of Townsend and Heuberger [47], and the control efficacy was evaluated using the Formula (2) of Abbott [48]:(1)$$P=\Sigma \left(n\times e\right)/\left(N\times E\right)\times 100$$
where *P* = percentage of damage; *n* = number of leaves for each level according to the scale; *e* = respective level of the scale; *N* = total number of leaves evaluated; and *E* = highest level of the scale.
(2)$$E_{C}=\left(C_{d}-T_{d}/C_{d}\right)\times 100$$
where *E_C_* = control efficacy, *C_d_* = severity in the control condition after application of the treatments, and *T_d_* = severity in the treatment condition after application.

### 4.4. Application of Chitosan for the Control of Powdery Mildew on Mixtecos Jalapeño Bell Pepper Plants under Greenhouse Conditions

#### Experiment 2

F1 Mixtecos jalapeño pepper plants were used with a powdery mildew disease severity of 2 to 4 according to the descriptive scale of the disease (Table 1) described by Guigón-López and González-González [2], with modifications from 12 to 16 levels. A completely randomized design with five replicates (one plant per replicate) per treatment was used: T1 = 0.0125% chitosan, T2 = 0.0025% chitosan, T3 = 0.05% chitosan, T4 = 0.1% chitosan, T5 = 0.2% chitosan, T6 = tebuconazole 25% (1.8 mL/L water), and T7 = control (water) (the same treatments described in experiment 1). Three applications of the treatments were carried out with intervals of 5 days between each one, and the percentage of incidence was evaluated by the number of diseased leaves per plant. In addition, the severity of the disease was evaluated 48 h after the last application [34]. The latter was conducted by comparison with the 16-level descriptive evaluation scale and subsequently the control efficacy was evaluated [49].

### 4.5. Data Analysis

Data obtained in both laboratory and greenhouse experiments were analyzed using a completely randomized design and subjected to an analysis of variance (ANOVA). Tukey’s means comparison test (*p* ≤ 0.05) was also used to analyze data from the laboratory experiment and Duncan’s means comparison test (*p* ≤ 0.05) was used for the analysis of the greenhouse experiment, both with the statistical program InfoStat version 2019.1.2.0.

## 5. Conclusions

*Leveillula taurica* was identified as the causal agent of powdery mildew on jalapeño pepper plants. The applications of treatments with 0.1 and 0.2% chitosan content were demonstrated to have a biocontrol effect on this disease under laboratory conditions, with a control efficacy of 57.70 and 65.39%, respectively. For its part, in plants grown in greenhouse conditions, the control efficacy of the treatments with 0.1 and 0.2% chitosan was 56.67 and 70.00%, these being superior to the other treatments, including tebuconazole chemical fungicide. It was also observed that this biopolymer affects the cell wall of the fungus. Therefore, chitosan could be considered as a bioalternative to the use of chemical fungicides for the control of *L*. *taurica*.

## Figures and Tables

**Figure 1 plants-13-00915-f001:**
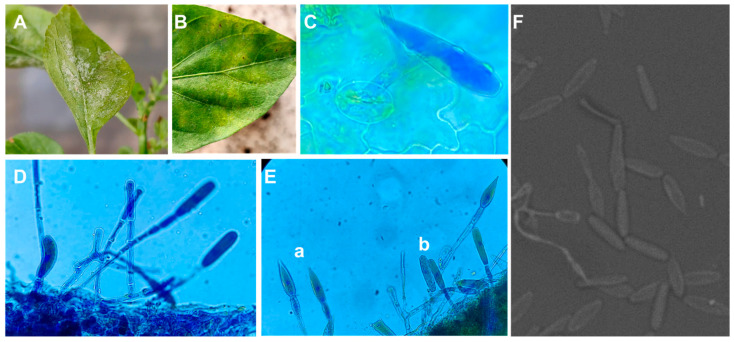
Morphological characteristics of *L. taurica*. (**A**) Whitish powdery mildew on the underside of leaves; (**B**) yellow chlorotic spots on leaf blades; (**C**) conidiophores emerging from leaf stomata; (**D**) branched conidiophores; (**E**) (**a**) lanceolate conidia and (**b**) cylindrical conidia; and (**F**) conidia visualized by scanning electron microscopy.

**Figure 2 plants-13-00915-f002:**
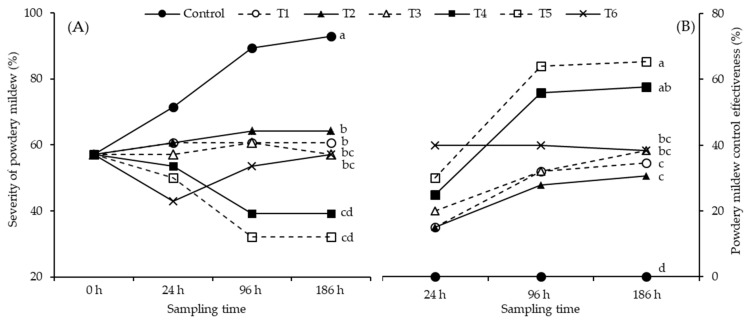
Application of different chitosan treatments (T1 = 0.0125% chitosan, T2 = 0.0025% chitosan, T3 = 0.05% chitosan, T4 = 0.1% chitosan, T5 = 0.2% chitosan, T6 = tebuconazole 25% (1.8 mL/L water), and T7 = control (water)), for the control of *Leveillula taurica* on Mixteco F1 jalapeño pepper. Different letters indicate differences between treatments (Tukey *p* ≤ 0.05). (**A**) Percentage severity of *L*. *taurica* and (**B**) percentage control efficacy on *L*. *taurica* powdery mildew.

**Figure 3 plants-13-00915-f003:**
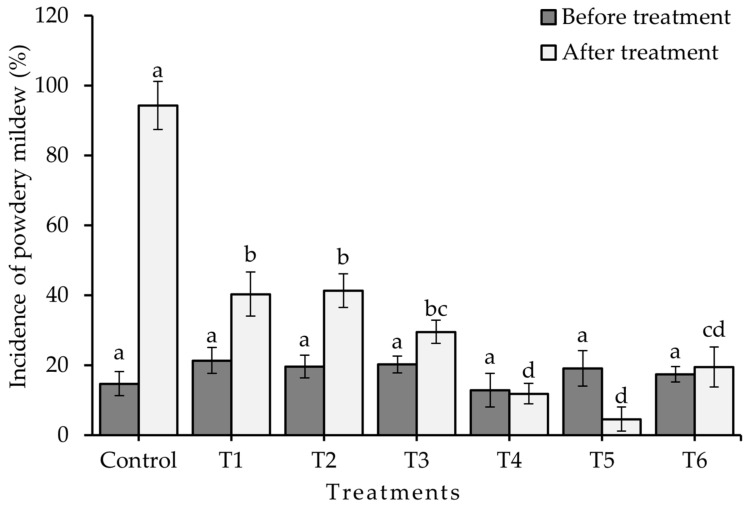
Reduction in the incidence of powdery mildew disease in the Mixteco F1 jalapeño pepper after the application of different concentrations of chitosan. ■ = incidence before treatment, □ = incidence after treatment. T1 = 0.0125% chitosan, T2 = 0.0025% chitosan, T3 = 0.05% chitosan, T4 = 0.1% chitosan, T5 = 0.2% chitosan, T6 = tebuconazole 25% (1.8 mL/L water), and T7 = control (water). Different letters indicate difference between treatments (Duncan *p* ≤ 0.05). *n* = 5 ± standard error.

**Figure 4 plants-13-00915-f004:**
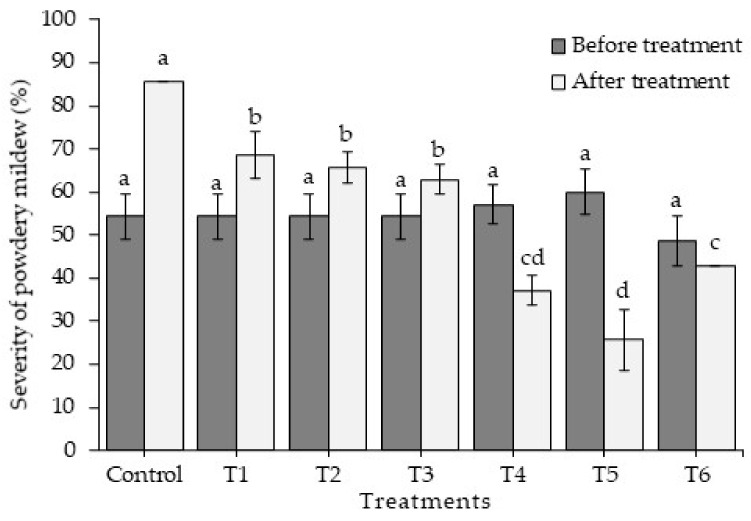
Reduction in the severity of powdery mildew disease in the Mixteco F1 jalapeño pepper after the application of different concentrations of chitosan. ■ = severity before treatment, □ = severity after treatment. T1 = 0.0125% chitosan, T2 = 0.0025% chitosan, T3 = 0.05% chitosan, T4 = 0.1% chitosan, T5 = 0.2% chitosan, T6 = tebuconazole 25% (1.8 mL/L water), and T7 = control (water). Different letters indicate difference between treatments (Duncan *p* ≤ 0.05). n = 5 ± standard error.

**Figure 5 plants-13-00915-f005:**
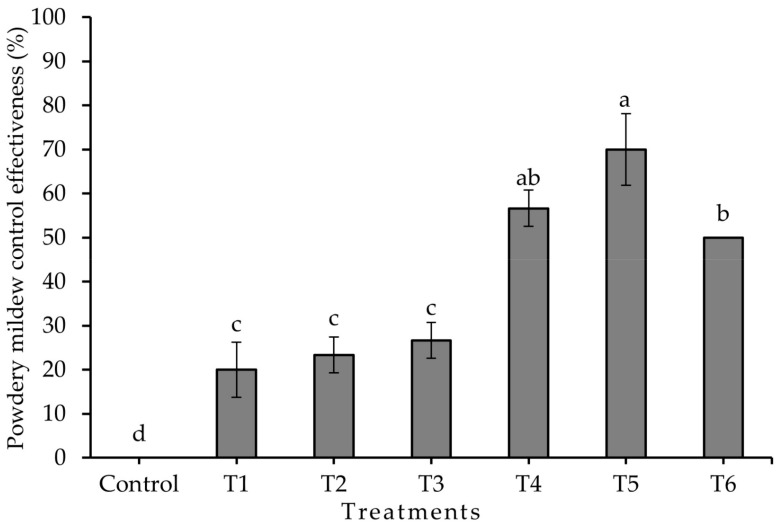
Control efficacy of different concentrations of chitosan on powdery mildew (*L*. *taurica*) on Mixteco F1 jalapeño pepper. T1 = 0.0125% chitosan, T2 = 0.0025% chitosan, T3 = 0.05% chitosan, T4 = 0.1% chitosan, T5 = 0.2% chitosan, T6 = tebuconazole 25% (1.8 mL/L water), and T7 = control (water). Different letters indicate difference between treatments (Duncan *p* ≤ 0.05). n = 5 ± standard error.

**Figure 6 plants-13-00915-f006:**
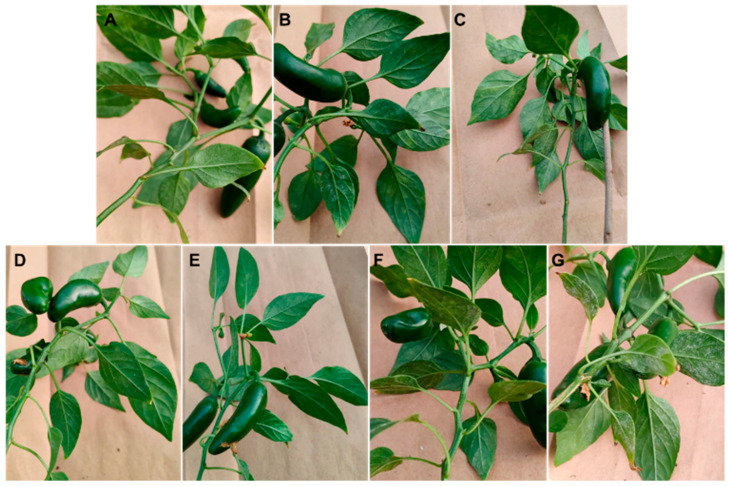
Effect of the application of treatments on Mixteco jalapeño pepper plants with the presence of *L*. *taurica*. (**A**) T1 = 0.0125% chitosan, (**B**) T2 = 0.0025% chitosan, (**C**) T3 = 0.05% chitosan, (**D**) T4 = 0.1% chitosan, (**E**) T5 = 0.2% chitosan, (**F**) T6 = tebuconazole 25% (1.8 mL/L water), and (**G**) T7 = control (water).

**Figure 7 plants-13-00915-f007:**
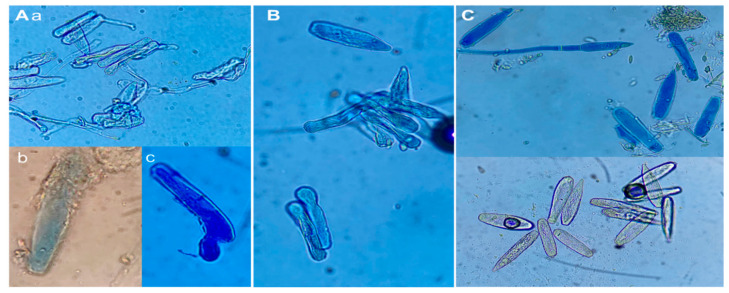
Effect of chitosan application on *Leveillula taurica*. (**A**) 0.2% chitosan application: (**a**) spores and mycelium with dry and deformed appearance, (**b**) spore with broken wall, and (**c**) deformed spore; (**B**) tebuconazole effect similar to that of chitosan; and (**C**) control treatment: spores with normal appearance.

**Figure 8 plants-13-00915-f008:**
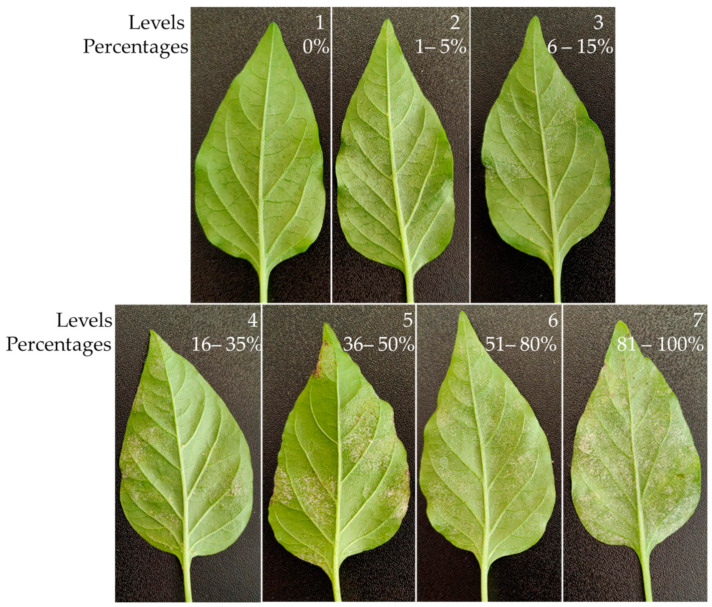
The visual scale of 7 levels of severity of powdery mildew on leaves of Mixteco F1 jalapeño pepper.

**Table 1 plants-13-00915-t001:** Descriptive scale for the evaluation of the severity of powdery mildew disease caused by *Leveillula taurica* on jalapeño pepper plants.

	% of Affected Area	
Levels	Plant	Leaf
1	0	0
2	1–30	1–15
3	1–30	16–35
4	1–30	36–50
5	1–30	51–80
6	1–30	81–100
7	31–60	1–15
8	31–60	16–35
9	31–60	36–50
10	31–60	51–80
11	31–60	81–100
12	61–100	1–15
13	61–100	16–35
14	61–100	36–50
15	61–100	51–80
16	61–100	81–100

## Data Availability

All data are contained within the manuscript.

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
