# Peer review of "Potential of Chitosan for the Control of Powdery Mildew (Leveillula taurica (Lév.) Arnaud) in a Jalapeño Pepper (Capsicum annuum L.) Cultivar"

_plants, 2024, doi:10.3390/plants13070915_

Round 1

Reviewer 1 Report

Comments and Suggestions for Authors

In the manuscript, Omar Jiménez-Pérez and colleagues investigated the potential of chitosan as an organic alternative to the use of fungicides for the control of powdery mildew in chili pepper.

However, this study suffers from a lack of novelty and research depth. I am uncertain how well this manuscript will appeal the general readership of Journal of Plants. Although the author' research focuses on the control of pepper powdery mildew using organic fungicides, I believe that this research topic already has a considerable amount of relevant literature and research findings in the current research field. Moreover, the content of the author's study is relatively simplistic and does not explore the mechanism of chitosan in controlling powdery mildew. I suggest that the authors could further deepen the research content in order to enhance the innovation and academic value of the study. I have some points for the authors to consider.

Major comments:

Chitosan can affect plant disease resistance through multiple mechanisms. It is suggested that the authors further investigate the mechanisms of chitosan in preventing powdery mildew in peppers from the following three aspects:

1. Promotion of pepper growth by chitosan.

2. Influence of chitosan on the structural disease resistance in peppers, such as thickening of plant cell walls or strengthening lignification.

3. Effects of chitosan on functional disease resistance in peppers, such as changes in enzyme activity related to antioxidant enzymes.

Furthermore, the authors are suggested to conduct research on the effects of chitosan in controlling powdery mildew under field conditions.

Minor comments:

Figure 3, 4: I suggest that the authors combine the two figures together, as they demonstrate similar conclusions. The same applies to the following figure as well.

Figure 6: Control and the T6 treatments did not show error bars in the data display.

Line 160: I would like the authors to provide a concise conclusion at the end.

Line 271: The authors state that “it does not cause damage to human health and the environment” However, “not cause damage” is not part of the study.

Reviewer 2 Report

Comments and Suggestions for Authors

- In the Introduction, line 84 must be changed, instead of "principal main" it is probably "principal aim"

-the paper shows the effect of chitosan in the control of Oidiopsis taurica in chili peppers, but it is not explained how the concentrations of chitosan used in the experiments were selected

- for accuracy and conclusive results, the experience should be repeated on a larger number of plants

- In figure 8, it is not specified what the images E), F) and G) represent

- In Discussions, line 261 is described as figure 9-D, but in figure 9 only A, B and C appear

Reviewer 3 Report

Comments and Suggestions for Authors

The manuscript describes interesting research for sustainable agriculture

Reviewer 4 Report

Comments and Suggestions for Authors

The current valid name of the pathogen is Leveillula taurica (Léveillé) G.Arnaud, Oidiopsis taurica is a synonym.

line 102: Oidiopsis taurica is not a genus but a species.

Inaccurate treatment of the referred sources: e.g. the “binding protein” or “78 kD” phrases can not be found at all in the referred article (15); de DOI and volume number of the same article are incorrect. Pls. check all the others!

“The control of cenicilla”: cenicilla is not ash but powdery mildew in English. Ash blight usually refers to Macrophomina phaseolina, a completely different pathogen.

Marking of the different treatments as ‘T1’ – ‘T6’ is misleading as ‘T’ suggest ‘time’. However, these are different concentrations of the chitosane. one treatment of tebuconazole and control.

Figure 2: the values presented on the two graphs are the reciprocal of each other: one of the two graphs (the second) is superfluous.

Figure 3: a percentage value higher the 100 on the vertical axis is meaningless. The legend of the figure suggest that powdery mildew infection was caused by the treatments. The same is valid for Figure 4.

Figure 5: The treatments are not continuous but discrete values so you can not present them with line diagram; use bar diagram instead. Again, as in the case of Fig. 2, these values are simply the reciprocals of those on Fig. 3. Fitting a regression on these values is pointless. (You could do it if you present the values only obtained with chitosan treatment.)

Discussion

It is stated, that the results of the presented work coincide with the results of others concerning the infection process of L. taurica.  However, this was not investigated (or not presented) in this study, so there is no coincidence.

The endoparasitic (and NOT endophytic) nature of the species L. taurica has nothing to do with the presented work so it is unnecessary to discuss. It is a mistake to list Phyllactinia as purely endoparasitic genus: it shows both ectoparasitic and endoparasitic growth. (And again, it has nothing to do with the presented work). Similarly, the detailed description of the conidiophores and conidium production of the genera Oidium, Oidiopsis and Ovulariopsis is superfluous, it can be found in any mycological text book.

Materials and methods

Differences in the Fig. 8 are hardly visible. “Manual atomizer” = Manual sprayer?

Comments on the Quality of English Language

The manuscript is characterised by too long, gramatically inaccurate, difficult-to-follow sentences and by incorrect usage of terms. I suggest to ask help from a native speaker or a professional translator.

Round 2

Reviewer 2 Report

Comments and Suggestions for Authors

Figure 1 - F). the image is black, nothing can be seen
